# Mineral Profile, Antioxidant, Anti-Inflammatory, Antibacterial, Anti-Urease and Anti-α-Amylase Activities of the Unripe Fruit Extracts of *Pistacia atlantica*

**DOI:** 10.3390/molecules28010349

**Published:** 2023-01-01

**Authors:** Mokhtar Benmohamed, Hamid Guenane, Mohammed Messaoudi, Wafa Zahnit, Chukwuebuka Egbuna, Majid Sharifi-Rad, Amina Chouh, Bachir Ben Seghir, Abdelkrim Rebiai, Sihem Boubekeur, Tarek Azli, Mohamed Harrat, Barbara Sawicka, Maria Atanassova, Mohamed Yousfi

**Affiliations:** 1Laboratory of Fundamental Sciences, University Amar Télidji of Laghouat, Road of Ghardaïa, Laghouat 03000, Algeria; 2Nuclear Research Centre of Birine, Ain Oussera, Djelfa 17200, Algeria; 3Chemistry Department, University of Hamma Lakhdar El-Oued, B.P.789, El Oued 39000, Algeria; 4Laboratoire de Valorisation et Promotion des Ressources Sahariennes (VPRS), Département de Chimie, Faculté des Mathématiques et des Sciences de la Matière, Université de Ouargla, Ouargla 30000, Algeria; 5Nutritional Biochemistry and Toxicology Unit, World Bank Africa Centre of Excellence, Centre for Public Health and Toxicological Research (ACE-PUTOR), Department of Biochemistry, University of Port-Harcourt, Rivers State 500001, Nigeria; 6Department of Range and Watershed Management, Faculty of Water and Soil, University of Zabol, Zabol 98613-35856, Iran; 7Laboratory of Microbiological Engineering and Application, Department of Biochemistry and Molecular and Cellular Biology, University of Mentouri Brothers Constantine 1, Constantine 25017, Algeria; 8Biotechnology Research Center CRBT, Constantine 25016, Algeria; 9Research and Development Centre RDC-SAIDAL, 35 Benyoucef Khattab Avenue, Mohammadi, El-Harrah, Algiers 16000, Algeria; 10Nuclear Research Centre of Draria, Algries 16050, Algeria; 11Department of Plant Production Technology and Commodities Science, University of Life Science in Lublin, Akademicka 15 Str., 20-950 Lublin, Poland; 12Nutritional Scientific Consulting, Chemical Engineering, University of Chemical Technology and Metalurgy, 1734 Sofia, Bulgaria

**Keywords:** *Pistacia atlantica*, unripe fruit, antioxidant activity, mineral, total phenol, biological activity

## Abstract

*Pistacia Atlantica* in folk medicine is used by Algerian traditional healers for treating a wide variety of diseases and conditions including dyspepsia, digestive problems, peptic ulcers, and, in particular, inflammatory diseases. The present study aimed to assess the phytochemical composition, in vitro antioxidant activity (using 2,2-diphenyl-1-picrylhydrazyl (DPPH), ABTS+, and reducing power methods), enzyme inhibitory activity (towards α-amylase and urease), antibacterial activity, and in vivo anti-inflammatory activity of the unripe fruit extracts of *Pistacia atlantica* collected from different parts of the Djelfa region of Algeria. According to the findings, various aqueous extracts exhibited significant antioxidant and enzymatic activities in all tests, but showed that they have a weak inhibitory effect against all tested bacterial strains. Twenty-one minerals comprising both macro- and microelements (Ba, Br, Ca, Cl, Co, Cr, Cs, Eu, Fe, K, Mg, Mn, Mo, Na, Rb, Sb, Sc, Sr, Th, U, and Zn) were determined using the technique of neutron activation analysis (INAA). The result indicates that the concentration of the mineral element is close to the minimal FAO recommendation. In addition, the result revealed significant anti-inflammatory activities. The data generated can be a valuable source of information for the pharmaceutical industry and medical research. These results suggest that the unripe fruit extracts of *Pistacia atlantica* have an appropriate potential to be utilized across a wide range of contexts as an agent with multifunctional uses, as well as a natural remedy for other physiological diseases.

## 1. Introduction

According to numerous reports, plants are a vast source of naturally occurring products, including the secondary metabolites that are the active ingredients in plants and are produced from the primary metabolic products, which are products necessary for human and animal body nutrition. In addition, plants use secondary metabolites as defense materials to protect themselves from any external attack [1,2,3,4]. Secondary metabolic products are important in promoting human health through the use of these products as antioxidant or antibacterial agents. Numerous secondary metabolic products with different chemical structures and groups, including phenols, have been shown to have broad biological effects in improving immune system effectiveness against viruses and other diseases that affect humans [5].

Phenolic compounds were initially investigated for their involvement in browning reactions and in providing a color and bitter taste; however, it was later shown that phenolic compounds protect plants against environmental factors such as low temperatures and free oxygen radicals [6]. More than 8000 different compounds are collectively referred to as phenolic compounds, making them the most significant class of phytochemicals [7,8], and the most prevalent in the plant kingdom [9]. Shahidi and Zhong [10] reported that phenolic compounds are ideal dietary antioxidants. In addition, they claimed that they have many health-promoting effects such as the stimulation of detoxifying enzymes and the inactivation of tumor-promoting enzymes. Again, recent scientific studies have shown that they have antibacterial, antifungal, anti-allergic, and anti-inflammatory activities, and other disease-mitigating properties against heart diseases, cancers, and neurological diseases [11,12].

Due to the importance of phenolic compounds, several studies have been conducted to investigate their chemistry in plant parts. In addition, research has been conducted to investigate the factors affecting the production of secondary metabolites [13,14]. Gairola et al. [15], studied the effect of aerial temperature changes on the content of bioactive compounds and reported that seasonal temperature variations had an impact on the amount of active compounds produced by plants, as there was an increase in the proportions of the chemical compounds with lower temperatures as a protective response from the plant [15]. In other studies, a strong correlation was found between the height above sea level and the amount of ultraviolet radiation, and the concentrations and amounts of chemical compounds including phenols for some plant species. These changes are responsible for the differences in the therapeutic properties of plants [16]. In addition, studies have been conducted on the effect of the degree of maturity on some plant parts such as seeds and the fruits of grapes [17], apples [18], kiwis [19], and pomegranates [20]. The results explained the variation in the quantities of phenolic compounds generated by plants throughout the growth stage since the content of phenolic compounds in the fruit decreases with the progression of ripening. This is true because certain immature fruits have more phenolic content than ripe ones [21].

Algeria has a rich and varied flora because of its geographical location and extensive and varied vegetation cover [22]. Among the flora, *Pistacia atlantica* commonly referred to as “Betoum”, is one of the many plants that grow spontaneously in Algeria and belongs to the *Anacardiaceae* family [23]. Different parts of *P. atlantica* (leaves, fruits, and aerial parts) are used extensively in food and traditional medicine, as well as alternative medicine to cure disorders of the digestive system, kidneys, heart, and respiratory systems, as well as to heal wounds [23,24]. *P. atlantica* has been the subject of numerous scientific investigations, which revealed its chemical composition in terms of its phenolic content, volatile components, fatty acids, tocopherols, and phytosterols compositions [25,26]. However, it has also been demonstrated that *P. atlantica* crude extracts have a range of pharmacological and biological properties, including antimicrobial, anti-inflammatory, analgesic, and antifungal properties. Others include anticarcinogenic, anticholinesterase, antidiabetic, and hepatic and renal stabilization properties. Other notable properties include wound healing, nipple cleft healing, and antispasmodic properties. The majority of these activities are attributed to the bioactive secondary metabolites of the plant, which include phenylethanoid glycosides, flavonoids, diterpenes, iridoids, and essential oil [27]. Given the importance of minerals and trace elements in numerous human metabolic processes [28,29,30], the micronutrients and potentially dangerous chemical elements were investigated using instrumented neutron activation analysis, one of the more delicate analytical techniques.

To the best of our knowledge, there is no previous study on the quantitative determination of phenolic contents and biological activities of the unripe fruit extracts of *P. atlantica*. Hence, this research aimed to assess the antioxidant, anti-α-amylase, anti-urease, antibacterial, and anti-inflammatory properties of the unripe fruit aqueous extracts of *P. atlantica*, as well as their total phenolic, flavonoid, and mineral contents.

## 2. Results

### 2.1. Mineral Elemental Analysis

In this work, the accuracy of the results and reliability and quality control of the INAA technique used in this study were checked, where the internal quality and quality control validation of analysis is a crucial step in the quality of results. In this investigation, we used the standard reference material of WEPAL-IPE 172 and WEPAL-IPE 189 as the standard, and used as the control samples.

From this analysis, Table 1 gives comparison results of the measured values with certified values in terms of the standard reference materials of IPE 189 and IPE 172 and the statistical testing Z-score and U-score values. It is observed that the Z-score values ranged from 0.03 to 2.65, and the U -score values ranged from 0.01 to 0.92. From these observations, this evaluation shows a good agreement and acceptable quality of the results obtained in this investigation.

In the present study, the mean values of mineral elemental concentrations were evaluated in the unripe fruit extracts of *Pistacia atlantica* obtained from different locations. These elemental contents measured by using the instrumental neutron activation analysis method are listed in Table 2, where (±) represent the uncertainties calculated.

The mean values of the mineral elements’ concentrations, their standard deviations, and the coefficients of variation assessed in the unripe fruits of *P. atlantica* are presented in Table 2.

Twenty-one elements viz (As, Ba, Br, Ca, Ce, Co, Cr, Cs, Eu, Fe, Hf, K, Na, Rb, Sb, Sc, Se, Sm, Sr, Yb, and Zn) were found in our samples conducted using INAA methods. The unripe fruits of *P. atlantica* produced the most calcium and potassium, and the least trace element europium (Eu) (Table 2).

Significant differences in the content of elements in the unripe fruit of *P. atlantica* between localities occurred in most cases. No significant differences in the content of elements between extractions were observed except in the case of Ce, Hf, Eu, and Sb. The coefficients of variation of the examined elements were generally very low or low. Europium (Eu) showed the greatest variability, while zinc (Zn) showed the greatest stability. The coefficient of variation informs us about the variability of the results, and observations about the “mean value”. This gives information about the scattering of the results, but in relation to how large the mean (median) is. This allows the determination of a relative measure of the scattering of results and makes it easier to compare the variability of given features among the same group of plants, in terms of the same feature. The coefficient of variation belongs to both classical and positional measures.

### 2.2. Assessment of Total Bioactive Compounds

Table 3 summarizes the spectrophotometric results of the bioactive components in the unripe fruit extracts of *P. atlantica*.

The highest total phenolic compounds content was found in the *P. atlantica* fruit extract obtained from TRC3, and the homologous content of these compounds was obtained for HCR1. The lowest content of them was recorded for SRC2 (Table 3).

The content of total phenolic compounds significantly depended on the extraction method of *P. atlantica* fruit. The BRC4 extract turned out to be the richest in phenolic compounds and the SRC2 extract turned out to be homogeneous in terms of the value of this feature. The TRC3 extract turned out to be the poorest in terms of flavonoids, and the HCR1 extract was homogeneous in this respect (Table 3).

### 2.3. Antioxidant Activity

The antioxidant properties of plant extracts will likely require investigation using many methods due to the complexity of phytochemicals. In this investigation, the assay models were practically related to the samples’ in vitro antioxidant activity. Aqueous extracts of the unripe fruit of *P. atlantica* from four different regions in Djelfa province (Algeria) were evaluated for their antioxidant activity using three different methods in this study, depending on several mechanisms of action, including DPPH scavenging activity, ABTS radical cation, and reducing capacity. The findings from these tests are shown in Table 4; the antioxidant activity of different aqueous extracts was evaluated using the A_0.5_ and IC_50_ indices.

In this study, the power reduction test method was employed. It is based on the idea that substances, in this case plant extracts that as expected have a reduction potential, react with potassium ferrocyanide (Fe^3+^) to form potassium ferrocyanide (Fe^2+^), which then reacts with chloride iron to form an iron–iron complex with a maximum absorption at 700 nm. Thus, power reduction was considered as the ability to donate electrons. Given the reducing agent in *P. atlantica* unripe fruit extracts, as is to be expected, the reducing force depend not only on the electron donor but also on the acceptor molecule. Ferric iron (Fe^+3^) is often used as an acceptor because it readily accepts electrons from almost all electron donors in biological systems. The A_0.5_ index was used to measure the antioxidant activity of the tested extracts. The highest value of this index was obtained in the standard object (BHT **). A significantly lower value was obtained for the TRC3 extract, and an even lower value in the case of the BRC4 extract, and the lowest values for the objects with the HCR1 and SRC2 extract; these values turned out to be homogeneous with the BHA ** standard object and ascorbic acid, used in this case as the reference object. The coefficient of variation for reducing power was low, which proves the high stability of the results obtained; the BHA standard object (Table 4) showed the greatest variability.

The results of the research on the possibility of using the DPPH radical to determine the antioxidant properties of the extracts obtained from the unripe fruits of *P. atlantica*, gave significant results. The highest IC_50_ value for DPPH was obtained in the standard (BHT **), and the lowest in the standard combination with BHA **. All the tested extracts turned out to be homogeneous in terms of this feature, which emphasizes the high and stable antioxidant potential of the tested compounds. The coefficient of variation, being a measure independent of the scale of units, allows for the assessment of the level of differentiation of the values of features in the population. The most stable IC_50_ value was obtained for the TRC3 extract, and the most variable was observed for the BHA standard. This allows us to define a relative measure of dispersion and makes it easier to compare the variability of given features in the same group, in terms of the same feature (Table 4).

The technique used to measure the antioxidant capacity of the extracts obtained from *P. atlantica*, in which the free radical is the ABTS compound, is an electron transfer method that also uses spectrophotometric detection to monitor the concentration of the colored radical cation ABTS. It shows four absorption maxima in the aquatic environment at the wavelengths 417, 645, 728, and 815 nm and three maxima in the ethanol environment—at 414, 730, and 873 nm. This technique made it possible to compare and evaluate the tested extracts in terms of their antioxidant capacity. The highest IC_50_ value expressed in μg/mL was obtained in three extracts: HCR1, SRC2, and TRC3, in relation to both standard objects and the BRC4 extract. The coefficients of variation for this feature were low, which means the high stability of the tested extracts. The object with the BRC4 extract turned out to be the most stable value of this feature (Table 4).

### 2.4. Anti-Inflammatory Activity

#### 2.4.1. In Vitro Anti-Inflammatory Activity

In vitro anti-inflammatory activity of the aqueous extract of aerial parts from four different locations of the unripe fruit of *P. atlantica*, was evaluated using a BSA denaturation assay. Table 5 shows that the plant extract exhibited a strong anti-inflammatory effect with BSA denaturation inhibitory potential estimated by 97.53% at 2 mg/mL. This was closer to the inhibitory potential exerted by the standard drug, diclofenac sodium, which showed 100% inhibition at 2 mg/mL.

#### 2.4.2. In Vivo Anti-Inflammatory Activity

The oral administration of the aqueous extract of aerial parts from four different locations of the unripe fruit of *P. atlantica* to mice with carrageenan-induced hind paw edema produced significant (*p* < 0.05) anti-inflammatory activity at the doses evaluated.

The present study shows that the aqueous extract of the unripe fruit of *P. atlantica* significantly reduced carrageenan-induced paw edema when compared to the reference drug, diclofenac sodium (10 mg/kg). The coefficients of the examined features were low, which means their high stability (Table 6).

### 2.5. Antimicrobial Susceptibility Assay

Antibacterial activity tests were performed on four Gram-positive and Gram-negative bacterial strains. The results of the evaluation of the antibacterial potential of the extracts are shown in Table 7. The antibacterial activity of the unripe fruit extracts of *Pistacia atlantica* were evaluated by the solid medium diffusion method using Mueller-Hinton medium for bacteria and Sabered medium for yeasts. The antibacterial activity was determined in terms of the diameter of the zone of inhibition (in mm) produced around the discs after incubation under suitable conditions for the development of the tested organisms.

The methods of extracting the unripe fruits of *P. atlantica* did not significantly differentiate the antibacterial activity carried out on four strains of Gram-positive and Gram-negative bacteria as well as on *Candida albicans* yeast. The coefficients of variation of the examined features were low, which proves it was a well-conducted experiment (Table 7).

### 2.6. Assessment of the Urease Inhibitory Activity

To evaluate the anti-urease activity of the unripe fruit extracts of *P. atlantica*, the effects of the extract on the urease enzyme were determined. From the results obtained (Table 8), the extracts of the unripe fruit of *P. atlantica* triggered an optimum response with a lower IC_50_ value than that of BRC3 and TRC4. The highest urease inhibitory activity was found in BRC3 (IC_50_ = 45.25 ± 1.07 µg/mL). This activity is higher than other extracts of *P. atlantica* and close to thiourea (IC_50_ = 11.57 µg/mL).

### 2.7. Assessment of α-Amylase Inhibitory Activity

From the results obtained (Table 9), the unripe fruit extracts of *P. atlantica* triggered an optimum response with a lower IC_50_ value than that of acarbose. The highest α–amylase inhibitory activity was found in TRC4 (IC_50_ = 32.62 ± 0.66). This activity is three hundred and thirty-three times (111) higher than acarbose (IC_50_ = 3650.93 ± 10.70 µg/mL). The α-amylase inhibitory activity of the remaining extracts turned out to be significantly higher than that of acarbose, but lower than that of TRC4, while the extracts: SRC1, SRC2, and BRC3 turned out to be homogeneous with regard to this feature (Table 9).

The coefficient of variation for the inhibitory properties of α -amylase was very low, which proves the high stability of the evaluated extracts and their inhibitory activity, where acarbose showed the greatest stability of the tested features (Table 9).

## 3. Discussion

There is convincing evidence, according to the WHO (World Health Organization), that mineral elements have biochemical, nutritional, and structural functions that are crucial for maintaining general physical and mental health [31,32]. Because of this, among the current objectives of this work, is to bridge the study gap using the INAA method to provide experimental evidence for the mineral composition of the unripe fruit of *P. atlantica*, along with their exact concentrations. On the other hand, before evaluating the use of pharmaceuticals, one of the most important components of safety is to monitor the mineral content of medicinal plants. *P. atlantica* is a well-known medicinal plant species that is widely utilized in folk and traditional medicine to provide sedative as well as spasmolytic effects. In the current investigation, the study focused on the unripe fruit of *P. atlantica*. The content of essential and toxic elements in the unripe fruit of *P. atlantica* from Algeria was examined using the INAA method.

The findings from this analytical study showed that there were twenty-one components, comprising macro and microelements, including Ba, Br, Ca, Cl, Co, Cr, Cs, Eu, Fe, K, Mg, Mn, Mo, Na, Rb, Sb, Sc, Sr, Th, U, and Zn. The results are presented in Table 2. The accuracy of these methods was verified and confirmed by analyzing several certified reference materials and the data are illustrated in Table 1.

The unripe fruit of *P. atlantica* collected from different parts of Djelfa province showed great variability in the concentration of macronutrients such as Ca, K, Fe, and Na. Calcium is a vital nutrient for human health. The intake of Ca is recommended to prevent osteoporosis and might be responsible for the absence of side effects regarding stomach lesions. The results obtained in this study show that Ca ranged from 1420 mg/Kg to 1712 mg/Kg. Both potassium (K) and sodium (Na), two macronutrients [33], are necessary for several biochemical reactions that take place in human body cells. These reactions minimize the risk of cardiac arrest, enhance heart function, boost metabolism, and regulate the body’s water balance. The concentrations of K ranged from 11,460 to 15,124 mg/kg, and Na ranged from 88 to 104 mg/kg, respectively. Iron is an essential element required for proper body functioning. The Fe concentration was found to range from 55 to 100 mg/kg in the studied sample, with the highest value of this element obtained in region 1 (HRC1). Micronutrients (Co, Cr, Se, and Zn) of the samples showed similarity in concentrations, except for cobalt in region 2 (SRC2), and Se in region 4 (BRC4) where the concentration was lower. It should be noted that samples from region 1(HRC1) always had the highest concentration except for zinc, where region 3 (TRC3) had the highest concentration.

According to Table 3, the range of total phenolic contents of aqueous extracts of immature fruit samples of *P. atlantica* varies by region between 213.70 and 227.03 mg GAE/g, with the highest value of phenolic content recorded in region 3 (TRC3) at 227.03 mg GAE/g, and the lowest value in the fourth region (SRC2) at 213.70 mg GAE/g, and the values in the first and second region being 219.29 and 223.41, respectively. The main observation from the results is that all studied samples contained a large amount of phenolic content. Based on these results found, the data obtained were higher than the results obtained in a study conducted by Benmahieddine et al. [34], who observed that the degree of fruit ripening affected the amount of phenols in the range of 10.11–64.14 mg/g. The study conducted by Belyagoubi, L [35], found that the solvents used for extraction influenced the phenolic content of ripe fruits, in the range of 2.30–31.86 mg/g. As for the aqueous extract, the result they obtained (129.16 mg/g) is lower compared to our result. However, for the methanolic extract, the amount of phenols was 285.95 mg/g, which is greater than our result, but our result was greater than the result reported by Toul et al. [36]. The effect of the solvent on the number of phenols was 205 mg/g. The study by Jouki from Iran [37] demonstrated the effects of two solvents (ethanol and methanol) on phenol contents. The result showed a phenolic content variation of 135.25 mg/g and 161 mg/g, respectively, which is also less compared to our results. Another study by Bakka et al. [38] determined the amount of phenols and flavonoids in the fruits and leaves of *P. atlantica*. They found that the amount of phenols and flavonoids in fruits was 12.97 mg/g and 0.825 mg/g, respectively, which are much lower than our results, but similar with the observation by Tahir et al. [39], who found the phenols and flavonoids content in fruits to be 5.20 mg/g and 1.01 mg/g, respectively. It should be noted that the results obtained in all studies revealed that the samples contained a significant amount of flavonoids compared to the amount of phenolic content. In contrast to the research we came across throughout our literature review, where the quantity of flavonoids was small in comparison to the amount of phenols, the number of flavonoids in this study makes up about half of the entire amount of phenols. The results showed that the aqueous extract of the unripe fruits is rich in phenolic content.

Antioxidant tests were conducted to assess the chelation, reduction, or inhibitory effects of the sample through indirect, direct, or competitive methods [40]. In the current study, the antioxidant activity of the unripe fruit of *P. atlantica* extracts was evaluated by DPPH, ABTS, and reducing power assays. The DPPH radical assay is the approach that is utilized the vast majority of the time to evaluate this antioxidant capacity. The unripe Fruit of *Pistacia atlantica* extracts and standard antioxidants (BHT and BHA) reduced the radical DPPH by combining H-atom and electron transfer processes, resulting in a transition from violet (DPPH^−^) to yellow (DPPH-H) at 515 nm. The capacity for reduction is measured by the degree to which antiradical drugs reduce absorption [41]. The results of DPPH scavenging (Table 4) showed that the unripe fruit of *P. atlantica* extracts exhibited a good antioxidant activity closer to that of BHT (IC_50_: 22.32 ± 1.19 μg/mL) and less than BHA (IC_50_: 5.73 ± 0.41 μg/mL). The DPPH radical scavenging activity of the different extracts is positively correlated with the TPC in the extracts. These results of DPPH^•^ scavenging are higher than previous studies by Benmahieddine and Hatamnia [34,42], which estimated the value in the range of (IC_50_: 38–2390 μg/mL). Moreover, when comparing the performance of region TRC3 with other extracts, it found that all aqueous extracts showed strong antioxidant potential utilizing DPPH radical scavenging activity, but were reported to be lower than our results [42,43]. The analyzed data of the ABTS assay showed that the BRC4 extract gave the best activity (IC_50_: 12.13 ± 0.16 μg/mL) compared with standards activity BHA and BHT acid (IC_50_: 1.81 ± 0.10 and 1.29 ± 0.30 μg/mL, respectively) followed by the HRC1, SRC2, and TRC3 extracts. Furthermore, BRC4 had a stronger scavenging action against the radical ABTS than it demonstrated against the radical DPPH. This can be explained by the difference in kinetics between the two reactions since the reduced form of ABTS is absent from the DPPH solution, while it is always present in the ABTS solution [44]. The results of the ABTS assay of our studies are better than that reported by Rigane et al. [42], who estimated this value in the range of ABTS (IC_50_ = 42 and 300 μg/mL). Moreover, these findings are higher than those of Peksel et al. [43] and Ghayth et al. [42]. In addition, the aqueous extracts of *P. atlantica* showed a remarkable ability to decrease iron ions, which may indicate their capability to regenerate oxidized molecules and break the hydrogen subtraction chain [45]. According to the data obtained from HCR1, the extract exhibited the best activity (IC_50_: 7.12 ± 1.66 μg/mL) very similar to α-tocopherol (IC_50_: 9.01 ± 1.46 μg/mL) and BHA (IC_50_: 8.41 ± 0.67 μg/mL). From the literature, our results are much better than those of previous works and estimated this value in the range of ABTS (IC_50_ = 50 and 6660 μg/mL) [46,47]. The antioxidant activity of phenolic compounds is mainly due to their redox properties, which can play an important role in the absorption and neutralization of free radicals, and the extinction of singlet, triplet oxygen, or peroxides in decomposition [48,49]. These natural compounds can reduce oxygen concentration and therefore exert their beneficial effects on health [50,51]. Indeed, it is well known that the consumption of fruit rich in phytochemicals such as polyphenols, carotenoids, and vitamins E and C reduce the risk of cancer and cardiovascular diseases [52,53].

On the other hand (in this study), the anti-inflammatory activities in vitro, and after in vivo, were carried out on Swiss albino mice through the inhibition of mouse paw edema induced by carrageenan. Based on the results obtained in vitro and in vivo anti-inflammatory testing; the anti-inflammatory activity extracts of the unripe fruits of *P. atlantica* from four different samples (Table 5 and Table 6) showed the value of the edema weight as a function of time in the mice compared to the control (diclofenac sodium), and treated batch, up to the end of four hours. On the other hand, oral administration of the aqueous extract effectively and dose-dependently prevented the development of edema, similar to the effects of diclofenac sodium (reference). The edema of the left hind paw was greatly reduced by the extract of the unripe fruits of *P. atlantica*. We observed an important reduction in the weight of the paw edema ranging from 26.30 to 79.50% for four samples, with the highest value in the sample recorded from region 32 (SRC2) when compared to the results of the reference diclofenac, with diclofenac sodium showing inhibition of 52.97%. Based on these results, it can be concluded that the unripe fruits of *P. atlantica* have potent anti-inflammatory activities. In addition to these pharmacological characteristics, the acute oral toxicity of extracts of the unripe fruits of *P. atlantica* was studied, with each group of mice receiving a single oral dose of 275 mg/kg, 1562 mg/kg, and 2555 mg/kg, respectively, of the aqueous extract of the unripe fruit; mice were observed daily for 14 days for behavioral changes, toxic reactions, and mortality, and the result was that this extract was non-toxic.

Urease (urea amidohydrolase) is present in various bacteria, fungi, algae, and plants. It is an enzyme that catalyzes the hydrolysis of urea into ammonia and carbamate in the last stage of nitrogen metabolism in living organisms [54]. Carbamate breaks down very quickly and spontaneously, resulting in a second ammonia molecule. These reactions can cause a significant increase in pH and are responsible for the negative effects of urease on human health but also have a negative effect on agriculture [55,56,57]. The results of our research (Table 8), showed significant inhibition of the urease of the tested extracts in relation to the total content of phenolic compounds and flavonoids, which depends on the method of obtaining the extract. It can be assumed that the flavonoids of the unripe fruits of *P. atlantica* are responsible for this urease inhibitory activity. However, further explanations and details regarding the isolation of the active ingredients from the plant may explain the chemical background of this phenomenon better. In terms of the inhibitory power, the content of flavonoids was followed by the water fraction and the crude extract of the plant, respectively. This reflects that the plant’s polar components were responsible for the current urease inhibitory action. Summing up, the unripe fruits of *P. atlantica* have strong urease-inhibiting components, which may be phenolic compounds (including flavonoids). Thus, the study provides scientific evidence for the traditional uses of this plant to treat stomach ulcers as well as other diseases by traditional methods.

Managing postprandial hyperglycemia by inhibiting carbohydrate hydrolase enzymes, primarily pancreatic amylase, is a key component of treating type 2 diabetes [58]. It catalyzes the hydrolysis of (1,4)D-glycosidic links of starch and certain other glucose polymers in patients with diabetes. Inhibiting amylase reduces starch breakdown, which leads to reduced blood glucose levels [59]. As shown in Table 9 the inhibition concentration of TRC4 (IC_50_ = 32.62 ± 0.66 μg/mL) against α-amylase was significantly almost 3 times higher than that exerted by SRC1 (IC_50_ value reached 110.71 ± 2.99 μg/mL) and even 325 times higher than that of acarbose, which was used as a positive control, while SRC2 and BRC3 extracts exhibited a strong inhibitory effect against α-amylase with the values IC_50_ = 69.22 ± 3.13 and 67.06 ± 1.74 μg/mL, respectively. Several factors, including geographic distribution and the impact of extreme climatic conditions (drought, irradiance, and salinity), which encourage the production of secondary metabolites such as polyphenols, could explain the observed variability in results. Previous research indicated that *P. atlantica* leaf aqueous extracts have inhibitory effectiveness on a level compared with that of a standard agent (acarbose) [60]. Moreover, Behmanesh et al. (2020) investigated the effects of oral administration of *P. atlantica* fruit extract on blood glucose levels and histomorphological modifications in the ovaries of rats with streptozotocin (STZ)-induced diabetes mellitus, demonstrating that the extract of *P. atlantica* fruits could be used in the treatment of rats with STZ-induced diabetes, with the objective of reducing ovarian difficulties [61]. Indeed, Ben Ahmed et al. (2018) studied the ability of *P. atlantica* Desf. leaves in inhibiting alpha-amylase depending on harvesting, month, and growing area. The extracts’ antidiabetic efficacy was stronger than that of acarbose [62]. Researchers have also shown that some phenolic acids and flavonoids have a significant inhibitory effect against alpha-amylase. Again, plant-based flavonoid catechin and myricetin (both constituents of *Pistacia atlantica*) are endogenous antidiabetic drugs with the advantageous ability to interact with enzyme–substrate complexes [63,64].

## 4. Material and Methods

### 4.1. Reagents and Materials

Folin–Ciocalteu (FCR), sodium carbonate (Na_2_CO_3_), 2,2-diphenyl-1-picrylhydrazyl (DPPH), 2,2′-azino-bis(3-ethylbenzothiazoline-6-sulfonic acid) ABTS, 2,6-di-tert-butyl-4-methylphenol (BHT), butylated hydroxyanisole (BHA), α-amylase from Aspergillus oryzae Green Alternative powder, ≥150 units/mg protein (biuret), gallic acid, quercetin, and AlCl_3_ were purchased from Sigma (Sigma-Aldrich, Taufkirchen, Germany). All the organic solvents and other chemicals used in the present study were of analytical grade and were obtained from Sigma-Aldrich (Sigma-Aldrich, Germany).

### 4.2. Plant Materials and Secondary Metabolites Extraction

The samples of the unripe fruit of *P. atlantica* were collected in August 2018, from four different parts of Djelfa province, Algeria; the first region is Hassi El Ouch, which is symbolized by (HRC1); (Latitude: 35°09′51.8″ N; Longitude: 3°16′19.5″ E; Altitude: 836 m). The second region is Bouira–Al Hadab (SRC2); (Latitude: 35°10′59.9″ N; Longitude: 2°55′13.5″ E, Altitude: 856 m). The third region is Touila-Had shary (TRC3); (Latitude: 35°27′20.8″ N; Longitude: 3°17′58.9″ E; Altitude: 735 m). The fourth region is Bousraya Ain Oussara (BRC4); (Latitude: 35°22′2.3″ N; Longitude: 2°56′27.0″ E; Altitude: 720 m), (Figure 1). The plant material was botanically identified and confirmed by Professor M. Yousfi, Laghouat University, Algeria. The unripe fruit of *P. atlantica* were dried in the shade and away from sunlight and moisture for 2 weeks. After that, the samples were ground to a fine powder using an agate mortar, and the pestle passed through a 120 µm mesh sieve and stored in airtight containers. The samples were subsequently extracted by cold maceration using water destitution, where 10 g of each sample was extracted with distillation water (100 mL, 2 day cycles, and three times), using an ultrasonic technic. The resulting suspension was filtered through Whatman filter paper (No. 1) and condensed in a vacuum evaporator at 40 °C.

### 4.3. Mineral Elements and Calculations

In this investigation, the mineral elements’ concentration in the unripe fruit of *P. atlantica* samples was determined using a sensitive nuclear analytical technique based on instrumental neutron activation analysis (INAA). Firstly, three samples of each collected charge weighing about 200 mg were stored in polyethylene-capped bottles. In addition, the accuracy of the method was evaluated by analyzing two certified reference materials (CRMs) IPE-172 and IPE-189 provided by IAEA. The samples and standard were packed and irradiated together for 4 h at the NUR research reactor with a thermal neutron flux of 4.5 × 10^13^ cm^−2^s^−1^. After appropriate cooling, the irradiated samples together with the standard were measured at different cooling times using a coaxial HPGe detector having the following characteristics: relative efficiency: 35%, FWHM 1.8 keV for the 1332.5 keV γ-peak of ^60^Co. In the end, the concentrations of minerals (major and trace elements) were determined using the equation of INAA as given by Equation (1) [65].
(1)$$\rho \left(µg/g\right)=\frac{\;{\left[\left(N_{p}/t_{c}\right)/DC\right]}_{a}}{{\left[\left(N_{p}/t_{c}\right)/DC\right]}_{s}}\frac{{\left(\rho W\right)}_{s}}{W_{a}}$$
where the subscripts *a* and *s* refer to the sample and the standard, respectively, *N_p_* is the net photo-peak counts, *W* is the sample mass, *D* = [exp (−λ t_d_)] is the decay factor, and *C* = ([1 − exp (−λ t_m_)]/λ t_m_) is the counting factor, *λ* the decay constant, t_c_, t_d_, and t_m_ the counting, decay, and measurement times, respectively.

To check the accuracy of the analytical method, the parameter of the U-score test was determined. They are calculated according to the following Equation (2):(2)$$U_{\mathrm{score}}=\frac{\left|X_{\mathrm{Lab}}-X_{\mathrm{Ref}}\right|}{\sqrt{\mu_{\mathrm{Lab}}^{2}+\sigma_{\mathrm{Ref}}^{2}}}$$
where X_Lab_, μ_Lab_, X_Ref_, and σ_Ref_ are the laboratory results, standard deviation, and the recommended and standard uncertainties, respectively. The laboratory performance is evaluated as satisfactory if U-score ≤ 1, and unsatisfactory for U-score ˃ 1 (result and certified value are not in agreement) [66].

### 4.4. Determination of the Total Phenols and Flavonoids Contents

#### 4.4.1. Total Phenol Content (TPC)

According to Müller’s microplate assay method, phenolics content was determined using the Folin–Ciocalteu reagent [67]. Briefly, 20 μL of the samples were added to 100 μL of Folin–Ciocalteu reagent diluted to (1:10) and 75 μL of sodium carbonate (Na_2_CO_3_) (7.5%). The resulting mixtures were incubated in dark condition at room temperature for 2 h. After that, their absorbance was measured at 765 nm. Total phenolic content was estimated operating the calibration curve of gallic acid (y = 0.0034x + 0.1044, R2 = 0.9972). The results were expressed in µg of gallic acid equivalent per milligram of extract (µg EAG/mg).

#### 4.4.2. Total Flavonoid Content (TFC)

Total flavonoid content (TFC) was estimated using the colorimetric procedure of Topçu et al. [68]. A volume of 50 µL of each extract was mixed with 130 µL of methanol, 10 µL of potassium acetate, and 10 µL of aluminum nitrate. The reaction mixture was incubated for 40 min in the dark at room temperature The absorbance of resulting mixtures was measured at 415 nm. Total flavonoid content was estimated operating the calibration curve of quercetin (y = 0.0048x, R2 = 0.997). The results are expressed in µg of quercetin equivalent per milligram of extract (μg EQ/mg of extract).

### 4.5. Evaluation of Biological Activity

#### 4.5.1. DPPH Scavenging Activity

Regarding the ability to donate hydrogen, the radical scavenging activity of the unripe fruit aqueous extracts of *P. atlantica* was evaluated using the conventional DPPH (1,1-diphenyl-2-picrylhydrazyl) method, which was followed by the procedure described by Blois [69]. First, the solution of DPPH was prepared in methanol (0.1 M), then 160 μL DPPH solution was added to 40 μL of different dilution extracts, samples, or synthetic antioxidants. After being shaken for 30 min in the dark, the resulting mixtures were kept at room temperature; the absorbances were measured at 517 nm using a microplate reader (Perkin Elmer EnSpire, New York, NY, USA). The synthetic antioxidants BHA and BHT were employed as a standard for comparison.

Percentage (%) DPPH free radical scavenging activity was calculated using Equation (3):I% (DPPH) = ((Ac − As)/Ac) × 100 (3)
where Ac is the absorbance of the DPPH solution and As is the absorbance of the sample. The results are expressed as IC_50_ values.

#### 4.5.2. ABTS Scavenging Activity

The ABTS scavenging activity of the unripe fruit aqueous extracts of *P. atlantica* was evaluated using spectrophotometry at varying concentrations against the ABTS cation radical (2,2’-azino-bis(3-ethylbenzothiazoline)-6-sulfonic) following the method described by Re et al. [70]. The ABTS^+^ was initially prepared in the following order: After dissolving 2 mM of ABTS in H_2_O and adding 2.45 mM of potassium persulfate (K_2_S_2_O_8_) to the solution, the combination was left to stand at room temperature and in the dark for 16 h. Then, 40 μL of each sample or standard drugs (BHT and BHA) prepared in methanol at various concentrations were mixed with 160 μL of diluted ABTS^•+^ solution (resulting in an absorbance value of 0.700 ± 0.025 at 734 nm). Equation (3) was used to determine the percentage of inhibition.

#### 4.5.3. Reducing Power (RP) Activity

The ability of *P. atlantica* extracts to reduce the ferric irons contained in the complex K_3_Fe(CN)_6_ to the ferrous irons was investigated, and the RP activity of the aqueous extracts was evaluated using the procedure outlined by Oyaizu [71], with some modifications. In brief, 10 μL of different concentrations of each extract or standard were added to a volume of 40 μL of sodium phosphate buffer (0.2 M, pH 6.6) and 50 μL of 1% K_3_Fe (CN)_6_ (potassium ferricyanide). Twenty minutes of incubation at 50 °C were provided for the reaction mixtures. The values obtained were presented as the concentration of the *P. atlantica* extract with an absorbance of 0.5 (A_0.5_).

#### 4.5.4. Anti-Inflammatory Activity

##### Evaluation of Possible Anti-Inflammatory Activity In Vitro

In this study, the test of In vitro anti-inflammatory activity was evaluated on the unripe fruit aqueous extracts of *P. atlantica*, by using the inhibiting denaturation of BSA (Bovine Serum Albumin) following the method previously reported by Kandikattu et al. [72] with slight modifications. It relied upon the inhibition of the denaturation of BSA caused by heat (72 °C). Briefly, 1 mL of each concentration of extract or standard (diclofenac sodium) was added to 1 mL of 0.2% BSA solution prepared in tris-HCl (pH = 6.6). The solutions were incubated at 37 °C for 15 min in an oven, then in a water bath at 72 °C for 5 min. After cooling, the turbidity was measured at 660 nm in a UV-visible spectrophotometer. For each concentration of extract, a blank is prepared in 1 mL of aqueous extract and 1 mL of tris-HCl (the purpose of this blank is to subtract the absorbance of the extract from the results obtained). The protective effect of samples against the denaturation of BSA is presented as an inhibition percentage calculated using the following formula: I% = [(Ac − As)/As] * 100, where I%: the inhibition percentage, AC: absorbance of the control, and AS: absorbance of the tested sample.

##### Evaluation of the Anti-Inflammatory Activity In Vivo

Both acute toxicity and anti-inflammatory studies were conducted on albino mice, weighing (20 to 25 g), of Swiss strain from the Pasteur Institute (Algiers). The laboratory animals were housed in plastic cages at room temperature (25 °C) and exposed to light for 12 h per day. During the acclimatization period (a week before being used in the various experiments), the mice had free access to water and food (croquettes from the Animal Feed Production Company, Bouzareah, Alger). The anti-inflammatory activity was assessed by inhibiting mouse paw edema induced by carrageenan following the protocol previously described by Colot [73]. The principle of this study consists of injecting carrageenin under the plantar fascia of the left paw of the mouse to cause an inflammatory reaction, which can be reduced by an anti-inflammatory product. This study allows the comparison of plantar edema after administration of the anti-inflammatory product to be evaluated (the aqueous extract of the plant at 10%) and the corresponding reference product (diclofenac sodium at 10 mg/kg). The experiment was conducted as follows:

At time T_0_, the mice were divided randomly into three batches; each batch contained five mice that were made up randomly. In the control group, each mouse received 0.5 mL of physiological water. Each mouse received a reference anti-inflammatory drug, diclofenac, in the standard group at 10 mg kg^−1^. The treated group received the test solution, where each mouse received 0.5 mL of the plant extract at 10%, where 5 g were suspended in 50 mL H_2_O to form a 10% tested solution. Before starting the experiments, the mice fasted for 16 h, were weighed, and then were administered intragastrical (gavage) for the three batches of these suspensions, namely, control solution, reference, and extract of the plant. Then half an hour after dosing, mice from the three batches received 0.025 mL of 1% carrageenan under the plantar fascia of the left hind paw of the mouse. Then four hours after this operation, the animals were sacrificed by cervical dislocation, the hind legs were cut at the height of the joint, and then weighed on an analytical balance. The anti-inflammatory activity was calculated as a percentage reduction in edema in the treated mice compared to the control according to the following formula:(4)$$I\%\;of\;edema=\frac{\left(P_{G}-P_{D}\right)\;control\;mouse)-\left(P_{G}-P_{D}\right)treated\;mouse{)}_{\;}}{\left(P_{G}-P_{D}\right)control\;mouse}$$
where *P_D_* = right leg weight and *P_G_* = left leg weight.

#### 4.5.5. Acute Oral Toxicity Test

In this study, acute oral toxicity occurs if adverse effects are observed after a single or multiple oral doses of the substance given over 24 h. This was carried out using OECD-423 guidelines [74]. Three groups of five mice each were prepared. They fasted 16 h before dosing with free access to water and then weighed. Each group, 1, 2, and 3, received orally a single dose of 275 mg/kg; 1562 mg/kg and 2555 mg/kg, respectively, of aqueous extract of the unripe fruit of *P. atlantica*, using a stainless steel cannula; then food was withheld for an additional 1 to 2 h. Mice were observed daily for 14 days for behavioral changes, toxic reactions, and mortality.

#### 4.5.6. Antibacterial Activity

To evaluate the antibacterial activity of the unripe fruit aqueous extracts of *P. atlantica*, the disc diffusion method was used. Five ATCC strains were evaluated, including *Staphylococcus aureus* (6538) and *Bacillus subtilis* (6633) (as Gram^+^ bacteria), *Pseudomonas aeruginosa* (9027) and *Escherichia coli* (8739) (as Gram^−^ bacteria), and *Candida albicans* (10231) (as Yeast). The paper disks of 6 mm diameter were impregnated with 35 µg of extract solution and then applied on the surface of the agar plates inoculated with microorganisms. Fosfomycin, carbenicillin, erythromycin, and cephalexin with 35 µg/disk were used as the reference standards to determine the sensitivity of Gram-positive and fosfomycin was used for Gram-negative bacteria strains. The plates were incubated at 37 °C to allow diffusion. The microbial strains and the culture media (Mueller-Hinton Agar for bacterial strains, Mueller-Hinton Agar supplemented with 2% glucose, and 500 mg/L Methylene Blue for fungal strains) were obtained from the Research and Development Centre (RDC SAIDAL), Algiers (Algeria). The culture media were kept at 37 °C for 48 h before use.

#### 4.5.7. Urease Inhibitory Assay

In brief, 10 µL of the extracts and the thiourea (standard) at various concentrations ranging from 3.125 to 100 µg/mL were combined in a 96-well microplate. The mixture was followed by adding twenty-five µL of urease solution (5 U/milliliter of urease from jack bean Canavalia ensiformis, type IX) and fifty microliters of urea solution (17 mM). In each well, 45 μL of phenol reagent (8% phenol and 0.1% *w*/*v* sodium nitroprusside) and 70 μL of alkaline reagent (2.85% NaOH and 4.7% active chloride NaOCl) were added following 50 min incubation at 30 °C. After this incubation, the absorbance was measured at 630 nm, and the result is reported as the IC_50_ value.

#### 4.5.8. α-Amylase Inhibition Assay

The α-amylase inhibitory activity was investigated according to Zengin et al. [58] using the iodine/potassium iodide method [58], with slight modifications. Briefly, the reaction mixture was prepared in a 96-well microplate by adding 25 μL of the sample at various concentrations with amylase solution in 1 U of sodium phosphate buffer (pH = 6.9 with 6 Mm NaCl). After incubating the resultant solution at 37 °C for 10 min, the reaction was started by adding fifty μL of a starch solution (1%). Simultaneously, a control was prepared without the enzyme solution. Re-incubation for 20 min at 37 °C was followed by the addition of HCl 25 μL (1 M) and 100 μL of iodine/potassium iodide solution to stop the reaction. The absorbance was measured at 630 nm, and the % inhibition of α-amylase was estimated as follows:(5)$$I\%=1-\frac{\left(Abs_{c}-Abs_{e}\right)-\left(Abs_{s}-Abs_{b}\right)}{\left(Abs_{c}-Abs_{e}\right)}$$

*Abs_s_* = absorbance (extract, starch, enzyme, IKI, HCl); *Abs_b_* = absorbance (extract, sodium phosphate buffer, IKI); *Abs_e_* = absorbance (solvent vol extract, enzyme, starch, HCl, IKI); *Abs_c_* = absorbance (solvent vol extract, sodium phosphate buffer, starch, HCl, IKI).

### 4.6. Statistical Calculations

The results obtained in this study were statistically analyzed using one-way ANOVA and, partially, correlation and regression analysis. The significance of the sources of variation was tested with the Fischer–Snedecor “F” test, and the significance of differences between the compared means was assessed using multiple Tukey intervals. Some results were subjected to multiple regression analysis. The function parameters were determined by the least squares method and the significance verification—by the Student’s *t*-test. T-Tukey’s multiple comparison tests enabled detailed comparative analyses of means by distinguishing statistically homogeneous groups of means (homogeneous groups) and determining the so-called least significant mean differences (NIR), which are marked by HSD (Tukey’s Honest Significant Difference) in Tukey’s tests. The ANOVA tables contain the most important elements of the analysis of variance, ending with the calculated probabilities (the so-called *p*-value) related to the applied test functions F (in the tables *p* = Pr. > F) (F Snedecor or Fisher–Snedecor). The calculated *p*-values determine the significance and size of the influence of the examined factors on the differentiation of the results of the analyzed variables by comparing them with the most commonly accepted significance levels *p* > 0.05. For detailed analyses based on t-Tukey’s multiple tests, the significance level was *p* = 0.05. In addition, descriptive statistics were performed in the SPSS and all other statistical analyses enabled by this program, including the coefficients of variation for the entire experiment (for each variable), CV (%), or RSD (relative standard deviation). They are measures of random variability in the conducted experiment.

## 5. Conclusions

This study reports on the mineral elements and some biological potentials of the unripe fruit of *Pistacia atlantica* L., obtained from four different parts of Djelfa region, Algeria. The data obtained in our study were very encouraging and showed that the unripe fruit extracts of *Pistacia Atlantica* have significant antioxidant and enzymatic activities in all tests but were shown to have a weak inhibitory effect against all tested bacterial strains. Moreover, the fruit can contribute to a healthier diet because it is rich in Ca, K, Fe, Se, and Na. The results of the present study also demonstrate that the aqueous extract of the unripe fruit of *P. atlantica* possesses significant anti-inflammatory activity on carrageenan-induced paw edema. The bioactive properties of the unripe fruit of *P. atlantica* make it a promising natural agent for future applications in the food and pharmaceutical industry.

Based on the results obtained from the unripe fruit of *Pistacia atlantica* L., aqueous extracts proved the presence of a variety of types of bioactive compounds that have interesting pharmacological effects. Furthermore, this investigation proved that the plant has distinctive medical, food, nutrition, and pharmaceutical properties. More studies are strongly recommended to establish the clinical significance of this plant in relation to the properties studied in this study.

## Figures and Tables

**Figure 1 molecules-28-00349-f001:**
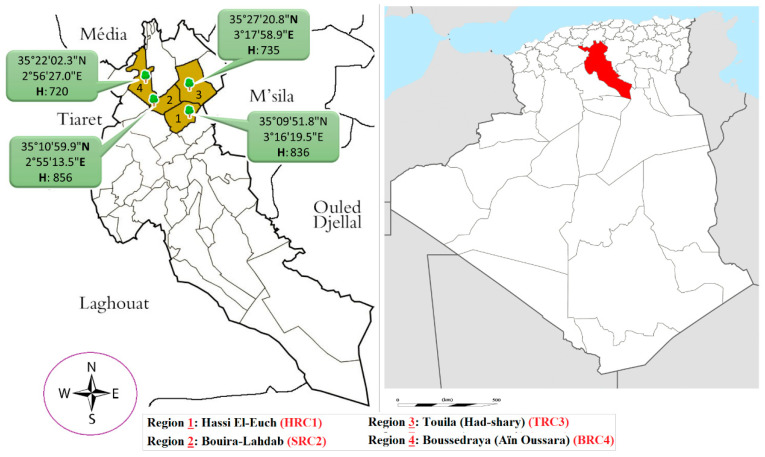
The geographical location of the sample collection.

**Table 1 molecules-28-00349-t001:** Comparison of measured mineral values with certified values in the standard reference materials of IPE 189 and IPE 172.

	Standards							
	IPE 189				IPE 172			
	Measured Values	Certified Values	U- Score	Z- Score	Measured Values	Certified Values	U- Score	Z- Score
Ba	74.52 ± 9.518	80.1 ± 5.96	0.5	0.94	17.62 ± 2.37	16.4 ± 0.99	0.48	1.24
Br	48.34 ± 29.64	40.6 ± 6.9	0.25	1.12	1.763 ± 0.361	2.1 ± 1.265	0.26	0.27
Ca	6346.5 ± 30.2	6730 ± 402	0.95	0.95	-	-	-	-
Co	0.0775 ± 0.0159	0.0789 ± 0.0132	0.06	0.1	17.62 ± 2.37	16.4 ± 0.99	0.48	1.24
Cr	0.919 ± 0.267	1.18 ± 0.228	0.74	1.14	1.763 ± 0.361	2.1 ± 1.265	0.26	0.27
Cs	0.0903 ± 0.0236	0.089 ± 0.0157	0.04	0.07	17.62 ± 2.37	16.4 ± 0.99	0.48	1.24
Fe	104.4 ± 12.8	110 ± 7.4	0.38	0.76	67.84 ± 6.51	64.4 ± 6.55	0.37	0.53
K	36,500.4 ± 4213.1	37,100 ± 2160	0.13	0.28	-	-	-	-
Rb	7.91 ± 0.59	7.5 ± 0.501	0.52	0.82	8.763 ± 1.167	9.07 ± 0.536	0.23	0.56
Sb	0.013 ± 0.007	0.0095 ± 0.0019	0.5	1.88	0.0372 ± 0.0105	0.0482 ± 0.0054	0.92	2.03
Sr	-	-	-	-	43.77 ± 3.14	42.2 ± 2.11	0.42	0.75
Zn	19.12 ± 1.39	20.1±1.32	0.51	0.74	33.21 ± 2.31	31.6 ± 2.19	0.51	0.74

(-): not reported.

**Table 2 molecules-28-00349-t002:** Content of elements in the unripe fruits of *P. atlantica* L. (mg/kg on a dry mass basis) determined by INAA technique and their variability.

Elements	The Unripe Fruits of *P. atlantica* *							
	HRC1		SRC2		TRC3		BRC4	
	Mean ± SD **	V ***	Mean ± SD	V	Mean ± SD	V ***	Mean ± SD	V ***
As	0.0103a,**** ± 0.0004	3.88	0.0097a ± 0.0003	3.09	0.0076b ± 0.0004	5.26	0.0090a ± 0.0002	2.22
Ba	0.63 ± 0.01	1.59	0.22c ± 0.002	0.91	0.33b ± 0.01	3.03	0.32b ± 0.01	3.13
Br	5.62b ± 0.112	1.99	6.58b ± 1.32	0.20	8.34a ± 0.17	2.04	8.08a ± 1.62	0.20
Ca	1 420.40b ± 28,408	2.00	1 626.51a ± 32.530	2.00	1 712.38a ± 34.25	2.00	1 633a.80 ± 32.68	2.00
Ce *	0.23a ± 0.003	1.30	0.22a ± 0.002	0.91	0.21a0 ± 0.003	1.43	0.20a ± 0.02	5.00
Co	0.051a ± 0.0088	1.71	0.0449a ± 0.0091	2.03	0.0194c ± 0.0006	3.09	0.0324b ± 0.0078	2.41
Cr	0.7184a ± 0.0145	2.02	0.4592b ± 0.00992	2.16	0.2969c ± 0.0071	2.39	0.2619c ± 0.00623	2.41
Cs	0.041b ± 0.0005	1.22	0.061a ± 0.004	6.56	0.064a ± 0.003	4.73	0.051a ± 0.002	3.92
Eu *	0.0009a ± 0.0001	11.11	0.0025a ± 0.0002	8.00	0.0008a ± 0.0001	12.50	0.0007a ± 0.0001	14.29
Fe	106.030a ± 2.976	2.81	69.49b ± 1.87	2.69	51.83c ± 1.42	2.74	54.35c ± 1.53	2.82
Hf *	0.022a ± 0.002	0.91	0.025a ± 0.002	8.00	0.027a ± 0.002	7.41	0.021a ± 0.001	4.76
K	11,460b ± 229	2.00	12,235b ± 245	2.00	15,124a ± 302.5	2.00	14,215a ± 284	2.00
Na	104.00a ± 2.08	2.00	102b ± 2.1	2.06	88.00b ± 1.76	2.00	98.00a ± 1.96	2.00
Rb	2.440a ± 0.038	1.56	3.44a ± 0.24	6.98	0.44b ± 0.02	4.55	1.44b ± 0.05	3.47
Sb	0.0030a ± 0.0001	3.33	0.0031a ± 0.0001	3.23	0.0038a ± 0.0001	2.63	0.0022b ± 0.0001	4.55
Sc	0.0036a ± 0.0004	1.11	0.0034a ± 0.0001	3.13	0.0043a ± 0.0001	2.33	0.0032a ± 0.0004	12.50
Se	0.0203b ± 0.00044	2.17	0.0197a ± 0.0013	6.60	0.0176a ± 0.0004	2.27	0.0090b ± 0.0011	1.22
Sm	0.0076a ± 0.00014	1.84	0.0074a ± 0.00012	1.62	0.0073a ± 0.0002	2.74	0.0072a ± 0.0007	9.72
Sr	5.9503b ± 0.4810	8.08	7.9403a ± 0.4810	6.06	0.9403a ± 0.0411	4.37	5.9403b ± 0.2210	3.72
Yb	0.670a ± 0.027	4.03	0.660a ± 0.011	1.67	0.771a ± 0.021	2.72	0.555b ± 0.011	1.98
Zn	10.108b ± 0.093	0.92	9.014ab ± 0.090	1.00	13.198a ± 0.117	0.89	10.919b ± 0.098	0.89

HRC1: means the first region is Hassi El Ouch; SRC2: means the second region is Bouira–Al Hadab; TCR3: means the third region is Touila-Had shary (TRC3); BRC4: means the fourth region is Bousraya Ain Oussara; * samples analysis; ** standard deviation; *** coefficient of variation (%); **** letter designations—the same letters mean that there are no differences between the considered averages.

**Table 3 molecules-28-00349-t003:** Total phenolics and flavonoids contents of the unripe fruit extracts of *P. atlantica*.

Extracts	Total Phenolic Compounds Content (mg GAE/mg) *	Flavonoids Content (mg QE/mg) **
HRC1 ***	223.4118a ± 1.4705	97.5000b ± 2.5043
SRC2 ***	213.7059c ± 0.5882	102.4306a ± 0.4419
TRC3 ***	227.0392a ± 1.6198	93.7500b ± 0.2946
BRC4 ***	219.2941b ± 1.4705	106.4583a ± 1.6204

Results are expressed as means ± SEM of three measures, * mg AGE/mg: milligram gallic acid equivalent/milligram of extract, ** mg QE/g: milligram quercetin equivalent/milligram of extract. HRC1 ***: means the first region is Hassi El Ouch; SRC2 ***: means the second region is Bouira–Al Hadab; TCR3 ***: means the third region is Touila-Had shary BRC4 ***: means the fourth region is Bousraya Ain Oussara.

**Table 4 molecules-28-00349-t004:** Antioxidant potentials of the unripe fruit extracts of *Pistacia atlantica* obtained from different locations.

Extracts	DPPH		ABTS		Reducing Power	
	IC_50_ (μg/mL)	V ***	IC_50_ (μg/mL)	V	A_0.5_ (μg/mL)	V
HRC1 *	16.30b ± 0.24	1.47	21.99a ± 1.25	5.68	7.12d ± 0.17	2.39
SRC2 *	17.33b ± 0.15	0.87	21.45a ± 0.55	2.56	9.48d ± 0.09	0.95
TRC3 *	16.28b ± 0.19	1.17	21.05a ± 0.19	0.90	28.54b ± 0.99	3.47
BRC4 *	16.63b ± 0.53	3.19	12.13b ± 0.16	1.32	15.07c ± 0.42	2.79
BHT **	22.32a ± 1.19	5.33	1.29c ± 0.03	2.33	50.1a ± 1.53	3.05
BHA **	5.73c ± 0.41	7.16	1.81c ± 0.10	5.52	8.41d ± 0.67	7.97
Ascorbic acid **	NT ****	=	NT	-	9.01d ± 0.15	1.66

A_0.5_: the concentration at the 0.50 absorption and IC_50_: the concentration at the fifty of inhibition. A_0.5_ and IC_50_ values represent the means ± SEM of three measures. Designations: ** standard compounds -; *** variability coefficients, **** NT: not evaluated. HRC1 *: means the first region is Hassi El Ouch; SRC2 *: means the second region is Bouira–Al Hadab TCR3 *: means the third region is Touila-Had shary; BRC4 *: means the fourth region is Bousraya Ain Oussara.

**Table 5 molecules-28-00349-t005:** In vitro anti-inflammatory activity of various concentrations (μg/mL) using BSA denaturation assay versus diclofenac sodium.

Concentration (μg/mL)	HRC1 *	SRC2 *	BRC3 *	TRC4 *	Diclofenac Sodium **
2000	nt	58.53 ± 3.06	55.00 ± 1.38	nt	100 ± 0.18
1000	nt	42.65 ± 0.81	50.65 ± 0.27	nt	92 ± 0.15
500	nt	24.88 ± 0.89	36.18 ± 0.44	nt	61 ± 0.15
250	nt	12.24 ± 0.37	26.76 ± 0.93	nt	37 ±0.18

HRC1 *: means the first region is Hassi El Ouch; SRC2 *: means the second region is Bouira–Al Hadab; TCR3 *: means the third region is Touila-Had shary; BRC4 *: means the fourth region is Bousraya Ain Oussara.; ** standard compounds, nt: not testing.

**Table 6 molecules-28-00349-t006:** The anti-inflammatory activity of the unripe fruit extracts of *Pistacia atlantica*.

Specification	Average Paw Weight (g)				% Edema	% Edema Reduction
	Left		Right			
	g	**** V (%)	G	**** V (%)		
HRC1 *	0.190b ± 0.001	0.526	0.142a ± 0.001	0.704	32.0%	26.30%
SRC2 *	0.135a ± 0.006	4.444	0.124b ± 0.003	2.419	8.9%	79.50%
BRC3 *	0.148a ± 0.009	6.081	0.129b ± 0.003	0.002	14.73%	66.01%
TRC4 *	0.155a ± 0.01	6.452	0.130b ± 0.006	4.615	19.23%	55.71%
Witness **	0.109c ± 0.006	5.505	0.076c ± 0.004	5.263	43.42%	0.0%
Diclofenac sodium ***	0.171b ± 0.009	5.263	0.142a ± 0.001	0.704	20.42%	52.97%

HRC1 *: means the first region is Hassi El Ouch; SRC2 *: means the second region is Bouira–Al Hadab; TCR3 *: means the third region is Touila-Had shary; BRC4 *: means the fourth region is Bousraya Ain Oussara. **: Physiological water. *** Standard compounds. **** V—coefficients’ variability (%).

**Table 7 molecules-28-00349-t007:** Antibacterial activity of the unripe fruit extracts of *P. atlantica* against bacteria and yeast strains.

Specification	Concentration (µL/disc)		(Diam) Diameter Inhibition (mm)					
			The Unripe Fruit of *P. atlantica*				Reference Drug	
	Groups		HRC1	SRC2	BRC3	TRC4		
Gram-positive	*Staphylococcus aureus* *ATCC 6538*	Diam Mean ± SD (mm)	9.0a ± 1.15	9.0a ± 1.15	9.0a ± 1.15	9.0a ± 1.15	fosfomycin 44 ± 0.5	carbenicillin 37.5 ± 0.4
		V *	12.78	12.78	12.78	12.78	1.14	1.07
	*Bacillus subtilis ATCC 6633*	Diam mean ±SD (mm)	9.0a ± 1.15	≥9	9.0a ± 1.15	9.0a ± 1.15	erythromycin 32.5 ± 0.3	cephalexin 31 ± 0.3
		V	12.78	-	12.78	12.78	0.92	0.97
Gram-negative	*Pseudomonas aeruginosa ATCC 9027*	Mean ±SD (mm)	9.0a ± 1.15	≥9	≥9	9.0a ± 1.15	fosfomycin 31 ± 0.21	
		V	12.78	-	-	12.78	0.45	
	*Escherichia coli ATCC 8739*	Diam Mean ±SD (mm)	9.0a ± 1.15	9.0a ± 1.15	9.0a ± 1.15	9.0a ± 1.15	fosfomycin 44 ± 0.31	
		V	12.78	12.78	12.78	12.78	0.70	
yeast	*Candida albicans ATCC 10231*	Mean ±SD (mm)	≥9	≥9	≥9	≥9	Nd	
		-	-	-	-	-	-	

V *—coefficients’ variability (%), Nd: not determined.

**Table 8 molecules-28-00349-t008:** The urease inhibitory activity of the unripe fruit extracts of *Pistacia atlantica*.

Extracts	Urease Inhibition (%)	
	IC_50_ (µg/mL)	V ***
HRC1 *	>100	-
SRC2 *	>100	-
BRC3 *	45.25c ± 1.07	2.36
TRC4 *	48.48b ± 7.52	2.21
Thiourea **	11.57d ± 0.68	5.88

HRC1 *: means the first region is Hassi El Ouch; SRC2 *: means the second region is Bouira–Al Hadab; TCR3 *: means the third region is Touila-Had shary; BRC4 *: means the fourth region is Bousraya Ain Oussara; ** standard compounds. *** V—coefficients’ variability (%).

**Table 9 molecules-28-00349-t009:** The α-amylase inhibitory of the unripe fruit extracts of *Pistacia atlantica*.

Extracts	Anti-*α*-Amylase	
	IC_50_ (µg/mL)	V ***
HRC1 *	110.71b ± 2.99	2.70
SRC2 *	69.22b ± 3.13	4.52
BRC3 *	67.06b ± 1.74	2.59
TRC4 *	32.62bc ± 0.66	2.02
Acarbose **	3650.93a ± 10.70	0.29

HRC1 *: means the first region is Hassi El Ouch; SRC2 *: means the second region is Bouira–Al Hadab; BCR3 *: means the third region is Touila-Had shary; BRC4 *: means the fourth region is Bousraya Ain Oussara.; ** standard compounds. *** V—coefficients’ variability (%).

## Data Availability

Data are available in the manuscript.

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
