# Peer review of "Mineral Profile, Antioxidant, Anti-Inflammatory, Antibacterial, Anti-Urease and Anti-α-Amylase Activities of the Unripe Fruit Extracts of *Pistacia atlantica"

_molecules, 2023, doi:10.3390/molecules28010349_

Round 1

Reviewer 1 Report

The study entitled “Mineral profile, Anti-Inflammatory and anti-Alpha-amylase, anti-Urease, Antioxidant and antibacterial activities of Unripe Fruit of Pistacia atlantica aqueous extract” is based on the primary and secondary phytochemical investigation of P. atlantica fruit, a dietary and medicinal plant. The manuscript has been written well. The introduction part is based on both old and new literature related to this work. The result and discussion part is comprehensive. The references provided fully support the given text. However, the authors are required to respond few concerns and bring about few minor changes/modification in this manuscript before the manuscript is being onward processed.

1.     Since there are many experimental constitution, more smooth storytelling is needed. Did the author emphasize the relationship between primary and secondary metabolites, if any, in the introductory part?

2.     In the manuscript, it is confusing whether the content emphasizes changes according to different regions or whether it is intended to show differences in compounds according to maturity. Changes in primary and secondary contents and biological activities according to maturity are mentioned very limitedly (e.g., amount of TPC), and even this is not a direct comparison between immature and mature fruits.

3.     It reflects various bioactive properties, but the experiment was designed only with aqueous extract of unripe fruit. Except for the content of TPC, most of the efficacy (anti-inflammatory, etc.) is due to the conviction that the ethanolic extract, in which secondary metabolites are intensively extracted, will have better efficacy.

Please provide clarification and emphasis on some questions.

Author Response

December 07, 2022

Journal: Molecules

Manuscript Number: Molecules-2079498

Mineral profile, Anti-Inflammatory and anti-Alpha-amylase, anti-Urease, Antioxidant and antibacterial activities of Unripe Fruit of Pistacia atlantica aqueous extract

Dear Editor           

We gratefully appreciate the rigorous review of the referees who reviewed our manuscript, which added much strength and validity to our research. Some points really helped to improve the manuscript and we really appreciate that. Again, we thank all the reviewers for their valuable inputs, which have given the manuscript a chance to reach a satisfactory level for publication.

(The corrected parts can be identified with the tracking in the revised manuscript)

Our responses are as follows:

Reviewer #1

Comments 1

The study entitled “Mineral profile, Anti-Inflammatory and anti-Alpha-amylase, anti-Urease, Antioxidant and antibacterial activities of Unripe Fruit of Pistacia atlantica aqueous extract” is based on the primary and secondary phytochemical investigation of P. atlantica fruit, a dietary and medicinal plant. The manuscript has been written well. The introduction part is based on both old and new literature related to this work. The result and discussion part is comprehensive. The references provided fully support the given text. However, the authors are required to respond few concerns and bring about few minor changes/modification in this manuscript before the manuscript is being onward processed.

Response 1

We would like to thank the reviewer for this comment and thorough reading of this manuscript and for the thoughtful comments and constructive suggestions.

Comments 2

Since there are many experimental constitution, more smooth storytelling is needed. Did the author emphasize the relationship between primary and secondary metabolites, if any, in the introductory part?

Response 2

Thank you for the comments. So, below is an attempt to clarify some of what I touched upon:

we have indicated in our introduction that, generally Medicinal plants are considered a huge factory for chemical compounds and are divided into two parts, primary and secondary metabolites, and we have said that the primary is responsible for growth and food, and the second is responsible for protection from external enemies.

Comments 3

In the manuscript, it is confusing whether the content emphasizes changes according to different regions or whether it is intended to show differences in compounds according to maturity. Changes in primary and secondary contents and biological activities according to maturity are mentioned very limitedly (e.g., amount of TPC), and even this is not a direct comparison between immature and mature fruits.

Response 3

We thank the reviewer for pointing this out, Dear reviewer, please let me give more clarification about our study,

Many studies have indicated that there is a significant relationship between the number of phenols and biological effects. Other studies have also indicated that the amount of phenols is affected by several natural factors, including the degree of ripeness and environment. in addition, many of them reported that the number of phenols decreases as maturity increases. as well as, through our research of the bibliography, we have found some that focus only on the part during and after maturity, and scarce of the Unripe part.

Base that. we have focused on the part of the Unripe Fruit of Pistacia Atlantica aqueous extract.

On the other hand, We also did not study the effect of ripening, but we did study the effect of changes in the areas.

Where our study showed that the amount of phenols is very large, compared to previous studies of fruits and even fruits of other species, even when they are extracted with methanol and ethanol. This is a positive result, so a large amount of phenols can be extracted using only water.

Comments 4

It reflects various bioactive properties, but the experiment was designed only with aqueous extract of unripe fruit. Except for the content of TPC, most of the efficacy (anti-inflammatory, etc.) is due to the conviction that the ethanolic extract, in which secondary metabolites are intensively extracted, will have better efficacy.

Response  4

We thank the reviewer for pointing this out, and we have agreed with the reviewer's comment.

But please let us clarify some think about our study.

Indeed, many studies have touched on the appropriate solvent for the extraction of phenols, and it has been found that ethanol and methanol give the maximum of yield.

In addition, we know that organic solvents such as methanol are toxic solutions and possibly carcinogenic

As for the water, it is easy to find, with much use in homes.

All these and also through our study, we have found that the amount of phenols is a significant amount. Where; this study showed that the amount of phenols is very important when compared to previous studies, even when extracted with methanol and ethanol.

in this work, when we study the aqueous extract, we found that it contains a large amount of phenols. Therefore, we wanted to study the biological effectiveness of the phenols extracted with water, and this is what we found that the aqueous extract has antioxidant activities, it was greater than the standard control used, and also we found alpha-amylase inhibition it is well.

These results that we reached are considered valuable results because we used only water in the extraction first for ease of obtaining and using it and avoiding organic solvents that may cause danger if they are not evaporated well. However, this research is part of what we are working on and we will discuss solvent extraction in later studies.

Comments 5

Please provide clarification and emphasis on some questions.

Response 5

Once again we thank the reviewer for this comment, thorough reading of this manuscript, and thoughtful comments and constructive suggestions.

We have tried hard to answer all the questions and clarify the ambiguity about the aim of our study

We hope the Reviewer and the Editors will be satisfied with our responses to the ‘comments’ and the revisions for the original manuscript.

 Thanks and Best Regards!

Yours Sincerely,

Reviewer 2 Report

In this work, the mineral elements and some biological potentials of unripe fruit of Pistacia atlantica L. were investigated. They found that the aqueous extracts had the highest antioxidant activities which may be due to the presence of phenolic compounds. The fruit can contribute to a healthier diet because it is rich in Ca, K, Fe, Se and Na. On the whole, the work is innovative, but there are many problems in writing, which need to be carefully revised.

1. There are too many grammar errors. They need to find a professional company to correct the language problem. For example: line 35, was should change to is; line 45, sentence rewriting; lines 664 669, on the other hand.

2. Many places are missing spaces, such as lines 156, 160, 270, 477.

3. In vitro” and “ P. atlantica” need italics. 50 in IC50 should be subscript. ml should be mL. It looks very non-standard overall.

4. The specific test methods are all reported methods. References are cited in the text, and specific steps can be put into SI.

5. The spectrogram data related to the test results should preferably be placed in SI. The liquid chromatographic data of the extract should be provided.

6. The format of references shall meet the requirements of journals. The journal name should be abbreviated.

Author Response

December 07, 2022

Journal: Molecules

Manuscript Number: Molecules-2079498

Mineral profile, Anti-Inflammatory and anti-Alpha-amylase, anti-Urease, Antioxidant and antibacterial activities of Unripe Fruit of Pistacia atlantica aqueous extract

Dear Editor           

We gratefully appreciate the rigorous review of the referees who reviewed our manuscript, which added much strength and validity to our research. Some points really helped to improve the manuscript and we really appreciate that. Again, we thank all the reviewers for their valuable inputs, which have given the manuscript a chance to reach a satisfactory level for publication.

(The corrected parts can be identified with the tracking in the revised manuscript)

Our responses are as follows:

Reviewer #2

Comments 1

In this work, the mineral elements and some biological potentials of unripe fruit of Pistacia atlantica L. were investigated. They found that the aqueous extracts had the highest antioxidant activities which may be due to the presence of phenolic compounds. The fruit can contribute to a healthier diet because it is rich in Ca, K, Fe, Se and Na. On the whole, the work is innovative, but there are many problems in writing, which need to be carefully revised

Response 1

We would like to thank the reviewer for this comment and thorough reading of this manuscript and for the thoughtful comments and constructive suggestions.

Comments 2

There are too many grammar errors. They need to find a professional company to correct the language problem. For example: line 35, was should change to is; line 45, sentence rewriting; lines 664 669, on the other hand.

Response 2

 Thank you so much for this careful and thorough comment; under the reviewer's suggestion, all typing and grammatical errors were corrected.

Comments 3

Many places are missing spaces, such as lines 156, 160, 270, 477.

Response 3     

We are really sorry for this typo error and it was corrected

Comments 4

“In vitro” and “ P. atlantica” need italics. 50 in IC50 should be subscript. “ml” should be “mL”. It looks very non-standard overall.

Response 4

 This was adjusted and tracked in the revised version of the manuscript.

Comments 5

The specific test methods are all reported methods. References are cited in the text, and specific steps can be put into SI.

Response 5

 Thank you so much for this careful and thorough comment; we have checked and corrected

Comments 6

The spectrogram data related to the test results should preferably be placed in SI. The liquid chromatographic data of the extract should be provided.

Response 6     

 Once again we thank the reviewer for this comment, thorough reading of this manuscript, and thoughtful comments and constructive suggestions.

Dear Editor, This work is considered the first of its kind to study the minerals present in the unripe Fruit of Pistacia Atlantica aqueous extract. as well as a study of some biological effects of aqueous extracts.

The results obtained showed that the aqueous extracts of the unripe fruits of Pistacia atlantica contain a huge amount of phenols and flavonoids, and they have great biological effectiveness as antioxidants, alpha-amylase inhibitors, kidneys, and anti-inflammatory agents, as well as non-toxicity, as shown in this study.

Also, the fruits are rich in healthy mineral elements, and the toxic elements are almost non-existent, as they were recorded in very small quantities, by using the INAA technique which is very known as an accurate technique.

All these results are what value ​​the use of this plant in traditional medicine.

As prospective studies, we will also work on future research on the use of organic solvents such as methanol, ethanol, and acetone in order to raise their efficiency in extraction. Also, we will work to identify the compounds responsible for the biological activity by means of high-quality liquid chromatography

Comments 7

The format of references shall meet the requirements of journals. The journal name should be abbreviated.

Response 7     

Thank you so much for this careful and thorough comment; we have checked and corrected

We hope the Reviewer and the Editors will be satisfied with our responses to the ‘comments’ and the revisions for the original manuscript.

 Thanks and Best Regards!

Yours Sincerely,

Reviewer 3 Report

Dear Authors,

My primary remark concerns your study design. You have applied in vivo studies for screening of anti-inflammatory activity, which can be done using multiple in vitro tests without harming animals. Firstly, you should use in vitro tests, ea. stabilization of human red blood cell (HRBC) membrane test or LPS-stimulated RAW 264.7 cells, and then if selected samples show promising activity go through to in vivo assays only with those samples. You have also not used any assay concerning in vitro cytotoxicity; instead, you’ve carried out in vivo toxicity evaluation. Bioethical considerations require compliance with 3R policies (Replacement, Reduction, Refinement) in animal experimentation, to avoid animal experiments altogether (Replacement), or to limit the number of animals (Reduction) and their suffering (Refinement) in tests to an absolute minimum. Additionally, every study involving animals requires the approval of the appropriate Bioethical Committee.

Antimicrobial assays based on agar diffusion methods are not acceptable in modern science. Please consider using dilution techniques and calculating MIC/MBC values. The disk diffusion method is commonly used to evaluate the sensitivity of clinically isolated microbial strains against antibiotics according to appropriate standards (e.g. CLSI). However, this method is unsuitable for assessing the antimicrobial potential of new molecules, plant extracts or fractions. Also, I suggest performing in vitro cytotoxicity evaluation to compare the result with antibacterial activity. This could allow for the assessment of selectivity.

Moreover, you should have performed a detailed phytochemical study.

Interestingly, there are 15 listed as authors of this manuscript, but in the Author contribution section, only three authors are listed as contributors. What did the rest of the co-authors do in this research?

Author Response

December 07, 2022

Journal: Molecules

Manuscript Number: Molecules-2079498

Mineral profile, Anti-Inflammatory and anti-Alpha-amylase, anti-Urease, Antioxidant and antibacterial activities of Unripe Fruit of Pistacia atlantica aqueous extract

Dear Editor           

We gratefully appreciate the rigorous review of the referees who reviewed our manuscript, which added much strength and validity to our research. Some points really helped to improve the manuscript and we really appreciate that. Again, we thank all the reviewers for their valuable inputs, which have given the manuscript a chance to reach a satisfactory level for publication.

(The corrected parts can be identified with the tracking in the revised manuscript)

Our responses are as follows:

Reviewer #3

Comments 1

Dear Authors, My primary remark concerns your study design. You have applied in vivo studies for screening of anti-inflammatory activity, which can be done using multiple in vitro tests without harming animals. Firstly, you should use in vitro tests, ea. stabilization of human red blood cell (HRBC) membrane test or LPS-stimulated RAW 264.7 cells, and then if selected samples show promising activity go through to in vivo assays only with those samples. You have also not used any assay concerning in vitro cytotoxicity; instead, you’ve carried out in vivo toxicity evaluation. Bioethical considerations require compliance with 3R policies (Replacement, Reduction, Refinement) in animal experimentation, to avoid animal experiments altogether (Replacement), or to limit the number of animals (Reduction) and their suffering (Refinement) in tests to an absolute minimum. Additionally, every study involving animals requires the approval of the appropriate Bioethical Committee.

Response 1

We would like to thank the reviewer for this comment and thorough reading of this manuscript and for the thoughtful comments and constructive suggestions, again thank you for all these valuable notes.

Yes, and as I mentioned, it was scheduled to conduct anti-inflammatory experiments in the laboratory ( In vitro) before testing them on animals, I'm so sorry , I forgot to put it in the in vitro anti-inflammatory results in the manuscript,, thank you for that note. (I added this result in the revised manuscript), where the test gives us a significant result

 But we did not experiment on animals until after we bibliography researched and in vitro testing this plant study and made sure that the parts of this plant have anti-inflammatory activity, especially the fruits.

All these, and in addition what encouraged us to test it on animals is the amount of flavonoids that we obtained, which is a very large quantity when compared to other studies on some plants that showed anti-inflammatory efficacy. For some samples, it is greater than the effectiveness of the drug used as a standard sample,

We will work in the next research to identify phenolic compounds by liquid chromatography in order to determine the responsible compound that has this effectiveness to a greater extent.

Comments 2

Antimicrobial assays based on agar diffusion methods are not acceptable in modern science. Please consider using dilution techniques and calculating MIC/MBC values. The disk diffusion method is commonly used to evaluate the sensitivity of clinically isolated microbial strains against antibiotics according to appropriate standards (e.g. CLSI). However, this method is unsuitable for assessing the antimicrobial potential of new molecules, plant extracts or fractions. Also, I suggest performing in vitro cytotoxicity evaluation to compare the result with antibacterial activity. This could allow for the assessment of selectivity.

Response 2

 It appears from the results obtained in this study, that the antibacterial activity of our samples studied in this experiment showed a very low sensitivity towards aqueous extracts, where the diameter of inhibition was equal to or less than 9 mm in some extracts, while others did not activity, produced no effect. These results show that the aqueous extracts have no sensitivity towards the bacteria studied, and as we initially used the highest concentration and its sensitivity was very low towards the bacteria, we have therefore not approached the study of the lowest slowed concentration of the bacteria, and we have just recorded the result that the aqueous extracts have no sensitivity towards the bacteria studied

Comments 3

Moreover, you should have performed a detailed phytochemical study.

Response 3     

  We thank the reviewer for this comment, thorough reading of this manuscript, and thoughtful comments and constructive suggestions.

Dear reviewer, This work is considered the first of its kind to study the minerals present in the unripe Fruit of Pistacia Atlantica aqueous extract. as well as a study of some biological effects of aqueous extracts. The results obtained showed that the aqueous extracts of the unripe fruits of Pistacia atlantica contain a huge amount of phenols and flavonoids, and they have great biological effectiveness as antioxidants, alpha-amylase inhibitors, kidneys, and anti-inflammatory agents, as well as non-toxicity, as shown in this study. Also, the fruits are rich in healthy mineral elements, and the toxic elements are almost non-existent, as they were recorded in very small quantities, by using the INAA technique which is very known as an accurate technique. All these results are what value ​​the use of this plant in traditional medicine.

As prospective studies, we will also work on future research on the use of organic solvents such as methanol, ethanol, and acetone in order to raise their efficiency in extraction. Also, we will work to identify the compounds responsible for the biological activity by means of high-quality liquid chromatography

Comments 4

Interestingly, there are 15 listed as authors of this manuscript, but in the Author contribution section, only three authors are listed as contributors. What did the rest of the co-authors do in this research?

Response 4

We are really sorry for this typo and it has been corrected. It is certain that all co-authors have participated in this research

We hope the Reviewer and the Editors will be satisfied with our responses to the ‘comments’ and the revisions for the original manuscript.

 Thanks and Best Regards!

Yours Sincerely,

Round 2

Reviewer 3 Report

Dear Authors,

Thank you for addressing my previous remark concerning the lack of compliance with 3R policies (Replacement, Reduction, Refinement) in animal experimentation. You have provided evidence of in vitro screening done before implementing animal studies. Please remember about this in future.

Concerning your explanation of the lack of MIC/MBC evaluation, I also agree that your extracts didn’t exert any antibacterial activity. However, in response to my remarks, you wrote: “we have just recorded the result that the aqueous extracts have no sensitivity towards the bacteria studied”, and this is in direct conflict with some statements found in the corrected manuscript: e.g. in the Abstract we read that “According to the findings, various aqueous extracts (…) showed a moderate inhibitory effect against all tested bacteria strains”. Confirming antibacterial activity of any kind requires performing MIC/MBC evaluation.  Please provide the necessary corrections.

Author Response

December 15, 2022

Journal: Molecules

Manuscript Number: Molecules-20794980 R1

Title: Mineral profile, Antioxidant, Anti-Inflammatory, Antibacterial, Anti-Urease and Anti-α-Amylase Activities of the Unripe Fruit Extracts of Pistacia atlantica

Dear Editor           

Dear Editor     

 Again, we sincerely thank the referees for their thorough examination of our work, which gave our research much-needed strength and authenticity. We greatly appreciate how much improvement the manuscript received from each point and remarks from respected reviewers. Again, we would want to express our gratitude to all of the reviewers for their insightful comments, which allowed the paper to improve to the point where it could be published.

(The corrected parts can be identified with the tracking in the revised manuscript)

Our responses are as follows:

Reviewer #3

Comments 1

Dear Authors,

Thank you for addressing my previous remark concerning the lack of compliance with 3R policies (Replacement, Reduction, Refinement) in animal experimentation. You have provided evidence of in vitro screening done before implementing animal studies. Please remember about this in future.

Response 1

We would like to thank the reviewer for this comment and thorough reading of this manuscript and for the thoughtful comments and constructive suggestions, once again thank you for all the valuable comments provided

Yes, based on the reviewer's suggestion for revision, we have made it, and in our future work, we will take their suggestions very carefully in our future work.

Comments 2

Concerning your explanation of the lack of MIC/MBC evaluation, I also agree that your extracts didn’t exert any antibacterial activity. However, in response to my remarks, you wrote: “we have just recorded the result that the aqueous extracts have no sensitivity towards the bacteria studied”, and this is in direct conflict with some statements found in the corrected manuscript: e.g. in the Abstract we read that “According to the findings, various aqueous extracts (…) showed a moderate inhibitory effect against all tested bacteria strains”. Confirming antibacterial activity of any kind requires performing MIC/MBC evaluation. 

Response 2

Again, thanks dear reviewer for this comment and thorough reading of this manuscript and for the thoughtful comments, valuable notes, and constructive suggestions.

We have agreed with her/his suggestion.

Based on the reviewer's suggestion we have checked and corrected these mistakes, and we have sorry about that.

We have mentioned in all manuscript revision version that aqueous extracts have a weak inhibitory effect against all tested bacterial strains.

Comments 3

Please provide the necessary corrections.

Response 3

 We are really sorry for these mistakes and it has been corrected. We thank the reviewer for pointing this out, and we have revised the manuscript according to the reviewer suggestions.

 We hope the Reviewer and the Editors will be satisfied with our responses to the ‘comments’ and the revisions for the original manuscript.

 Thanks and Best Regards!

Yours Sincerely,